# Safety of Bedside Portable Low-Field Brain MRI in ECMO Patients Supported on Intra-Aortic Balloon Pump

**DOI:** 10.3390/diagnostics12112871

**Published:** 2022-11-19

**Authors:** Christopher Wilcox, Matthew Acton, Hannah Rando, Steven Keller, Haris I. Sair, Ifeanyi Chinedozi, John Pitts, Bo Soo Kim, Glenn Whitman, Sung Min Cho

**Affiliations:** 1Division of Cardiac Surgery, Department of Surgery, Johns Hopkins University School of Medicine, Baltimore, MD 21287, USA; 2Division of Critical Care, Department of Medicine, Mercy Hospital of Buffalo, Buffalo, NY 14220, USA; 3Department of Biomedical Engineering, Johns Hopkins University School of Medicine, Baltimore, MD 21287, USA; 4Division of Thoracic Surgery, Department of Surgery, Johns Hopkins University School of Medicine, Baltimore, MD 21287, USA; 5Division of Neuroradiology, Department of Radiology and Radiological Science, Johns Hopkins University School of Medicine, Baltimore, MD 21287, USA; 6Hyperfine, Inc., Guilford, CT 06437, USA; 7Neuroscience Critical Care Division, Departments of Neurology, Neurosurgery, Anesthesiology and Critical Care Medicine, Johns Hopkins University School of Medicine, Baltimore, MD 21287, USA

**Keywords:** portable MRI, ECMO, VA ECMO, IABP, safety, brain injury, neuroimaging

## Abstract

(1) Background: Fifty percent of patients supported on veno-arterial extracorporeal membrane oxygenation (VA-ECMO) are concurrently supported with an intra-aortic balloon pump (IABP). Acute brain injury (ABI) is a devastating complication related to ECMO and IABP use. The standard of care for ABI diagnosis requires transport to a head CT (HCT) scanner. Recent data suggest that point-of-care (POC) magnetic resonance imaging (MRI) is safe and may be effective in diagnosing ABI in ECMO patients; however, no data exist in patients supported on ECMO with an IABP. We report pre-clinical safety data and a case series to evaluate the safety and feasibility of POC brain MRI in ECMO patients supported with IABP. (2) Methods: Prior to patient use, ex vivo testing with an IABP catheter within the Swoop^®^ Portable MRI (0.064 T) System™ was conducted. After IRB approval, clinical testing was performed for the safety and feasibility of early ABI detection. (3) Results: No deflection force was measured with a 7.5 French Maquet Linear IABP within the 0.064 T field. Three adult ECMO patients (average age: 40 years; 67% female) supported with IABP completed four POC brain MRI exams (median exam time: 30 min). Multiple signal abnormalities were detected on the POC brain MRI, corresponding to HCT results. (4) Conclusions: Our preliminary results suggest that adult VA-ECMO patients with IABP support can be safely imaged with low-field POC brain MRI in the intensive care unit, allowing for the early and bedside imaging of patients.

## 1. Introduction

The use of veno-arterial extracorporeal membrane oxygenation (VA-ECMO) has rapidly expanded over the past decade for patients in cardiogenic shock [1]. Although VA-ECMO has the ability to provide full circulatory support, it can compromise the left ventricular (LV) function and exacerbate cardiac failure [2]. To circumvent the adverse effects of VA-ECMO, a concomitant mechanical circulatory support (MCS) device can be placed to unload the LV and reduce afterload to prevent worsening myocardial function, decrease the myocardial oxygen demand, and enhance circulatory support [2]. The most common choice for an adjunctive MCS device is an intra-aortic balloon pump (IABP), which is used in 50% of all VA-ECMO patients [3]. IABP decreases the cardiac afterload, with a subsequent enhancement in diastolic coronary blood flow [2].

Although MCS devices can provide circulatory support with potential outcome benefits, neurologic complications are common in patients with MCS [4,5]. Additionally, the transport of critically ill patients and poor sensitivity of head CT (HCT) for ischemic injury remain significant obstacles in making a timely and accurate diagnosis of an acute brain injury (ABI) in patients supported with MCS. Magnetic resonance imaging (MRI) remains the gold standard for diagnosing ABI; however, conventional systems utilize high-strength magnetic fields (1.5–3 T) that are incompatible with MCS devices such as ECMO and IABP circuits.

Recent advances in low-field (0.064 T), portable MRI technology enable the acquisition of clinically meaningful imaging in the presence of ferromagnetic materials. We have previously demonstrated the safety and feasibility of low-field point-of-care (POC) MRI to obtain brain imaging in ECMO patients [6]. However, there are no safety data on POC MRI in ECMO patients who are supported on IABP. Historically, IABP has been regarded as a contraindication to MRI due to the electrical equipment necessary for the IABP to function [7]. However, there has been no published evidence to support this claim.

The purpose of the study is to evaluate the safety and feasibility of low-field POC brain MRI in ECMO patients supported on IABP with both pre-clinical and human research.

## 2. Materials and Methods

Clinical data were acquired from patients enrolled in an ongoing prospective observational study of POC MRI in adults with veno-arterial or veno-venous (VA- or VV-) ECMO that was approved by the institutional review board of Johns Hopkins Hospital (IRB00285716). Written consent was obtained and documented from a legally authorized representative as ECMO patients were unable to provide consent.

### 2.1. Pre-Clinical Safety Assessment

Prior to clinical implementation, a preliminary risk assessment was performed using a 7.5 French Maquet Linear IABP catheter with the Swoop^®^ Portable MR Imaging System™ (MK1.6 and MK1.7, Hyperfine, Inc., Guilford, CT, USA). Preliminary testing was performed to evaluate whether the catheter was deflected due to the magnetic field of the Swoop system. This was performed by constructing a nonmagnetic mechanical fixture, which included a spring gauge to measure the displacement of the catheter. The spring gauge was connected to the tip of the catheter via a spring-and-pulley system. The mechanical fixture and catheter were placed on the patient bridge and slowly moved into the MRI coil. At this point, the spring gauge was used to measure the deflection of the magnetic force on the catheter.

The primary outcome for the pre-clinical testing was to determine the deflection force induced by the magnetic field on the IABP catheter. According to the ASTM F2052-15 standards, a device is MR-compatible if the magnetically induced deflection force is less than the force on the device due to gravity. If the deflection force exerted on the IABP catheter was greater than the force of gravity, the study would not progress to the clinical phase.

### 2.2. POC Brain MRI

Exclusion criteria included weight over 200 kg, pregnant patients, and contraindications to 1.5 T MRI other than ECMO and IABP. Incompatible implants included left ventricular assist devices, pacemakers, Impella^®^ devices, and any metallic foreign bodies within the chest.

The study procedures were previously reported in our prior study [6]. A POC brain MRI was obtained using the 64 mT Swoop^®^ MR imaging system (MK 1.6, Hyperfine, Inc., Guilford, CT, USA). The MR system was wheeled into the patient’s room with all ICU equipment outside the magnet’s 5-Gauss line. Once the patient’s head of bed was aligned with the head coil, 4 trained individuals, including a perfusionist, respiratory therapist, intensivist, and nurse, slid the patient into position using a lift-and-slide maneuver. The patient was maintained as flat as possible while positioning at the direction of the team leader. Pads were placed around the patient’s head to prevent motion (Figure 1). A physician monitored vital signs, ECMO flow, and the positioning of the cannulae, and the endotracheal tube continuously during the exam. The following changes were considered serious adverse events (SAEs): (i) change in mean arterial pressure (MAP) of ±20%, (ii) decrease in ECMO flow rate of 10%, or (iii) decrease in oxygen level (SpO_2_) of 10% from baseline. Resultant images were read by a neuroradiologist (H.I.S.) who was blinded to clinical information.

Primary outcomes for the clinical study were safety and feasibility, defined as completion of the POC MRI exam without SAEs. The secondary outcome was the clinical result of MR images compared to HCT images.

## 3. Results

### 3.1. Pre-Clinical Safety Assessment

Zero deflection force was measured when a 7.5 French Maquet Linear IABP catheter was placed within the 0.064 T field (Figure 2).

### 3.2. POC Brain MRI

We report three patients (average age: 40 years; 67% female) that underwent a total of four POC brain MRI exams while on concomitant VA-ECMO and IABP support. Two patients (67%) had post-cardiotomy shock, and one (33%) had acute cardiogenic shock secondary to COVID-19 myocarditis (Table 1). Two (67%) had IABP placed one day prior to ECMO cannulation; one (33%) had IABP and VA-ECMO placed concurrently. The median time between the start of IABP support and POC brain MRI was three days. The median number of days of IABP and combined VA-ECMO/IABP support were 3 and 7, respectively. MRI exams were completed in 28, 29, 30, and 46 min, respectively.

### 3.3. Clinical Presentation and Imaging Findings

Patient 1: A 72-year-old male with a past medical history of hypertension, diabetes, and end-stage renal disease that was cannulated on VA-ECMO via the right atrium and ascending aorta for post-cardiotomy shock. His cardiac function initially improved, and he was transitioned from VA-ECMO to IABP support after two days of VA-ECMO support. However, his cardiac function quickly deteriorated solely on IABP support, and he was placed back on VA-ECMO support in addition to IABP. He underwent his first HCT on day 2 of IABP (day 1 of ECMO) and was found to have global hypoxic ischemic brain injury and cerebral edema. A second HCT performed on day 4 of IABP (day 3 ECMO) had no changes from the prior HCT. POC MRI exams were performed immediately after each HCT, and both showed T2 signal abnormalities within the occipital lobes and left temporal and parietal lobes. For each POC MRI, the IABP was placed on a 1:2 rate, triggered with EKG waveform.Patient 2: A 37-year-old female with a past medical history of asthma, who was cannulated on VA-ECMO via the femoral vein and artery with IABP support for cardiogenic shock secondary to COVID-19 myocarditis. She underwent a POC brain MRI on day 2 of IABP (day 1 of ECMO), which showed focal encephalomalacia of the left cerebellar hemisphere. She did not undergo HCT imaging while on VA-ECMO and IABP support. During the POC MRI, the IABP was placed on a 1:1 rate, triggered with arterial pressure.Patient 3: A 71-year-old female with a past medical history of hypertension, diabetes, and chronic kidney disease who was cannulated on VA-ECMO via femoral vein and artery and IABP for post-cardiotomy shock. She underwent an HCT on day 1 of IABP (day 1 of ECMO), which was significant for age-indeterminate infarcts in the right thalamus and the right parietal and left occipital lobes. A subsequent POC brain MRI on day 4 of IABP (day 4 of ECMO) was without abnormalities. During the POC MRI, the IABP was paced on a 1:1 rate, triggered with EKG waveform.

Figure 3 shows sample image slices of different MRI sequences available and a comparison of HCT performed generally within 24 h; however, in the case of Patient 2, it was not performed until after ECMO decannulation.

## 4. Discussion

We previously reported that adult patients supported on VA- and VV-ECMO can be safely imaged with a low-field POC brain MRI at the bedside in an ICU setting [6]. However, a significant proportion of ECMO patients were not enrolled in our study due to the concurrent use of IABP with no available safety data. Pre-clinical testing of the IABP device provided conclusive evidence that there would be no deflection or movement within the 5-Gauss line. This result provided confidence that there would be no disruption of patient hemodynamics, allowing us to include the device in our clinical study.

We report here the first use of POC brain MRI for patients supported on both IABP and VA-ECMO and for a patient centrally cannulated on VA-ECMO. We successfully conducted POC brain MRI without SAEs, demonstrating the safety and feasibility of a POC brain MRI for ECMO patients supported with IABP. Additionally, no ABI were detected on POC brain MRI imaging, corresponding to clinical HCT results. In our previous report, we collected four MR sequences: T1-weighted, T2-weighted, fluid-attenuated inversion recovery (FLAIR), and diffusion-weighted imaging with an apparent diffusion coefficient map (DWI ADC), which resulted in an average exam time of 39 minutes [6]. In this study, the T1 sequence was eliminated, resulting in an average run time of 33 min, without a compromise in diagnostic ability.

As part of the preparation for MRI, the IABP was switched from an EKG trigger to a pressure trigger. If the patient was unable to pressure-trigger the IABP, EKG leads were moved to the patient’s hips to be outside the 5-Gauss line, and proper EKG triggering was confirmed after positioning into the MRI. There was minor EKG interference during the DWI and ADC sequences that was not sustained and had no effects on he0modynamics, ECMO flow, or IABP inflation.

As this is the first ever series of MR images acquired on ECMO patients with IABP, there were several challenges encountered in this early experience. In order to have ideal MR brain images, the head needed to be within 4 cm of the MRI coil. Patients 1 and 3 had a large body habitus (BMI = 44.7 and 44.8, respectively) that prevented their heads from being placed within four centimeters of the MRI coil, resulting in partial brain coverage. In Patient 2, MR images were collected while she contained metallic pins within her hair, resulting in significant artifacts on the MR images. Lastly, Patient 3 had significant use of accessory muscles for breathing during the scan, resulting in significant head motion. While pads were placed along the side of the head to minimize the head motion, the MRI had to recalibrate motion sensing, in addition to the DWI ADC sequence requiring a second run to collect images. A larger head coil to accommodate obese patients would improve image acquisition in all patients. Further investigation with the adjustments above is needed to determine if the image quality can be improved to the point that ABI can be reliably detected.

Overall, it is notable that POC MRI had no impact on the hemodynamic status (ECMO flow and MAP) and oxygenation in all patients with IABP support. Therefore, we demonstrated that POC brain MRI in an ECMO patient supported with IABP is safe and offers logistic advantages over conventional neuroimaging. Accessible POC brain MRI has the potential to markedly improve our ability to diagnose subclinical ABI and to immediately alter clinical management with the goal of mitigating injury in a patient population that had previously been excluded.

## 5. Conclusions

Adult ECMO patients with IABP support can be safely imaged with a low-field POC brain MR at the bedside in an ICU setting, which may allow the early detection and timely intervention of ABI.

## Figures and Tables

**Figure 1 diagnostics-12-02871-f001:**
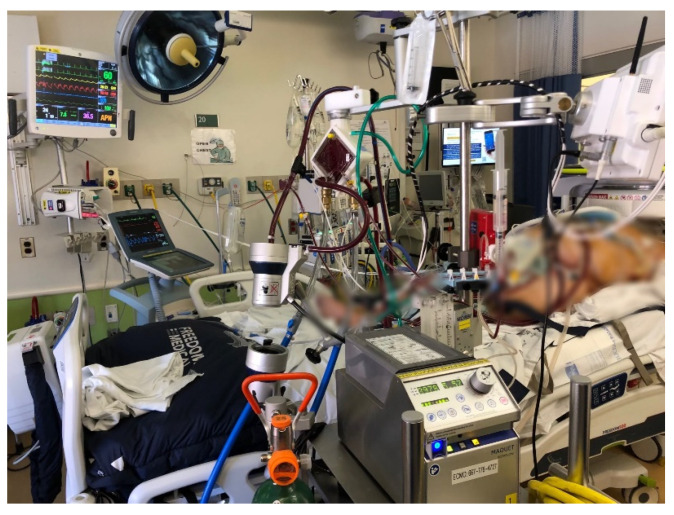
Patient with ECMO support within the Swoop^®^ scanner. ECMO and IABP console were kept outside the scanner 5-Gauss line. * ECMO: extracorporeal membrane oxygenation, IABP: intra-aortic balloon pump, 5-Gauss: typical safety line around a magnetic resonance imaging device outside of which functioning of medical and other electronic devices is tested.

**Figure 2 diagnostics-12-02871-f002:**
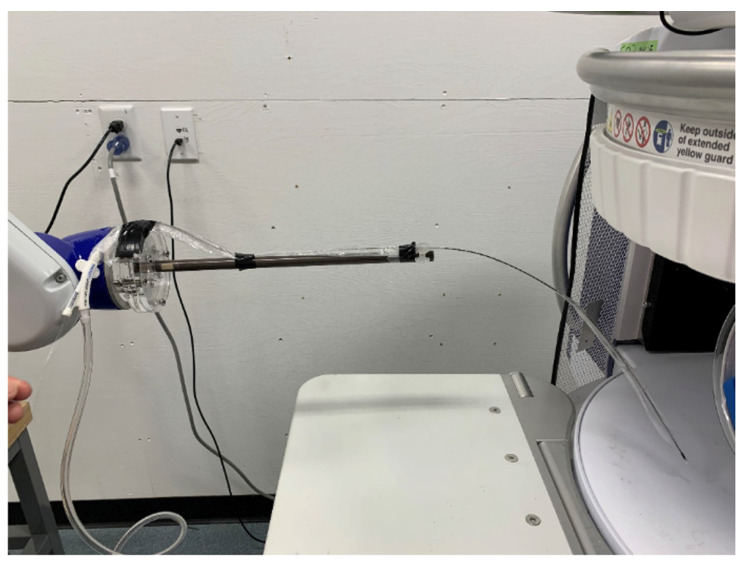
The 7.5 Fr Maquet IABP catheter in the patient opening of the low-field POC MRI system. IABP: intra-aortic balloon pump, MRI: magnetic resonance imaging, POC: point-of-care.

**Figure 3 diagnostics-12-02871-f003:**
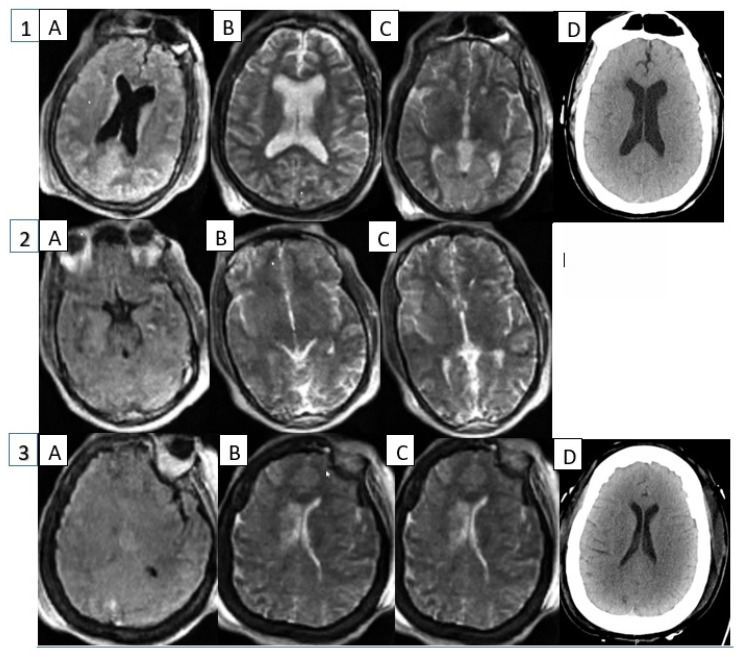
Point-of-care MRI and comparative HCT images. Representative point-of-care MRI images of three critically ill patients with an intra-aortic balloon pump (IABP). Panel (**A**) represents flair images, (**B**) and (**C**) are T2 sequences, and (**D**) represents comparative head computed tomography (CT) image when available. Patient 1 demonstrated periventricular white matter hypo-intensity concerning for chronic ischemic changes. Patient 2 demonstrated left focal cerebellar encephalomalacia with a right sided artifact from an ultrasound probe. No abnormalities were discovered in Patient 3.

**Table 1 diagnostics-12-02871-t001:** Patient characteristics and adverse events during portable brain MRI scan.

MRI Study	Patient	Age (yrs)	Sex	BMI	ECMO Indication	Cannulation Strategy	Neurologic Symptoms	MRI Time	HCT Finding	MRI Finding	Adverse Events
**1**	1	72	Male	44.7	Post-Cardiotomy Shock	Right Atrium, Aorta (V-A)	Coma under sedation	28 min	No acute findings	T2 signal abnormality in occipital lobes and left temporal and parietal lobe	None
**2**	1	72	Male	44.7	Post-Cardiotomy Shock	Right Atrium, Aorta (V-A)	Coma under sedation	29 min	No acute findings	occipital and left parietal and temporal lobe T2 abnormality	None
**3**	2	37	Female	29.1	COVID-19 myocarditis	Fem-Fem (V-A)	Coma under sedation	32 min	Not applicable	encephalomalacia left cerebellar hemisphere	None
**4**	3	71	Female	44.8	Post-Cardiotomy Shock	Fem-Fem (V-A)	Coma under sedation	46 min	Small hypoattenuating lesions in right thalamus, parietal and left occipital lobes, indeterminate	No acute findings	None

## Data Availability

Data supporting results can be obtained through contact with the corresponding author. As data involve human subjects at Johns Hopkins Hospital, appropriate offices will need to notified prior to data sharing.

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
