# Peer review of "Safety of Bedside Portable Low-Field Brain MRI in ECMO Patients Supported on Intra-Aortic Balloon Pump"

_diagnostics, 2022, doi:10.3390/diagnostics12112871_

Round 1
Reviewer 1 Report
The authors introduced the use of POC-MRI in three patients with ECMO and IABP support, emphasizing that POC-MRI does not have a significant impact on the function of IABP, and it could complete brain examinations without affecting the patients' hemodynamics and MCS function.
Compared with the authors' previous studies (Cho SM, Wilcox C, Keller S, Acton M, Rando H, Etchill E, Giuliano K, Bush EL, Sair HI, Pitts J, Kim BS, Whitman G. Assessing the SAfety and FEasibility of bedside portable low-field brain Magnetic Resonance Imaging in patients on ECMO (SAFE-MRI ECMO study): study protocol and first case series experience. Crit Care. 2022 Apr 30;26(1):119), the difference is that the patients in this study have contemporary IABP support. In order to minimize the possible interference of POC-MRI on IABP, the authors used Linear IABP instead of Sensation Plus IABP, and used a pressure trigger to avoid possible interference of EKG. I appreciate the authors' efforts for ABI in MCS patients.
I have three suggestions that the author should make some improvements.
The first is the reference. I don’t know why the first item in the reference list is missing, and reference #6, which has been cited many times in the article, seems to be reference #7?
Second, the comparison image of POC-MRI and HCT in this paper showed relatively poor quality (Figure 3), and I suggest the authors improve the picture quality.
In addition, as far as Figure 3 is concerned, compared with HCT, POC-MRI does not have much advantage in image quality. Whether subclinical ABI can really be detected effectively requires continuous quality improvement in POC-MRI and more clinical cases to prove. This issue should be addressed in the discussion.
Author Response
Reviewer 1
The authors introduced the use of POC-MRI in three patients with ECMO and IABP support, emphasizing that POC-MRI does not have a significant impact on the function of IABP, and it could complete brain examinations without affecting the patients' hemodynamics and MCS function.
Compared with the authors' previous studies (Cho SM, Wilcox C, Keller S, Acton M, Rando H, Etchill E, Giuliano K, Bush EL, Sair HI, Pitts J, Kim BS, Whitman G. Assessing the SAfety and FEasibility of bedside portable low-field brain Magnetic Resonance Imaging in patients on ECMO (SAFE-MRI ECMO study): study protocol and first case series experience. Crit Care. 2022 Apr 30;26(1):119), the difference is that the patients in this study have contemporary IABP support. In order to minimize the possible interference of POC-MRI on IABP, the authors used Linear IABP instead of Sensation Plus IABP, and used a pressure trigger to avoid possible interference of EKG. I appreciate the authors' efforts for ABI in MCS patients.
I have three suggestions that the author should make some improvements.
The first is the reference. I don’t know why the first item in the reference list is missing, and reference #6, which has been cited many times in the article, seems to be reference #7?
Thank you for pointing this out. We have checked and this was a copy/paste issue into the template that we did not catch. As there was no reference listed under the number ‘1’ all of the subsequent references have a numerical value 1 higher than intended. This has been corrected and rearranged in the manuscript.
Second, the comparison image of POC-MRI and HCT in this paper showed relatively poor quality (Figure 3), and I suggest the authors improve the picture quality.
Thank you for the suggestion, we have recreated the images and attached into the manuscript file.
In addition, as far as Figure 3 is concerned, compared with HCT, POC-MRI does not have much advantage in image quality. Whether subclinical ABI can really be detected effectively requires continuous quality improvement in POC-MRI and more clinical cases to prove. This issue should be addressed in the discussion.
Thank you for your candor here regarding quality. A summary sentence ‘Further investigation with the adjustments above is needed to determine if the image quality can be improved to the point that ABI can be reliably detected.’ Was added as a conclusion to the 4th paragraph of the discussion where we discuss limitations affecting image quality and areas for future improvement. Overall, the MRI device is capable of improved image quality – in these initial scans the priority was on safety and team coordination/communication. We learned a substantial amount regarding ways to optimize positioning and reduce ‘noise’ or interface in the images. Our goal and expectation is that the quality of the images should come in line with what the Swoop® portable MRI device has demonstrated in the general population as our technique and positioning improve (and perhaps with some device specific modifications for a severely immobile population).
Reviewer 2 Report
This study investigated the feasibility of bedside portable low-field brain MRI in extracorporeal membrane oxygenation (ECMO) patients supported on intra-aortic balloon pump. Three patients were involved.
1. This study is actually a case report. Does this special issue of Diagnostics support such an article type?
2. The article type in Page 1 ‘Research Manuscript’ would be better to change to something like ‘Case Report.’
3. ‘Recent data suggests point-of-care (POC) magnetic resonance imaging (MRI) is safe and effective in diagnosing ABI in ECMO patients, no evidence exists regarding the safety of MRI for IABP patients.’ This sentence should be revised.
4. There are ‘5G’ and ‘5-gauss’ and ‘5 Gauss’ throughout the manuscript. Please unify the descriptions. In addition, ‘5G’ may be mistaken to be understood as ‘5-Generation.’
5. ‘Impella devices’ would be better to change to ‘Impella® devices’?
6. Figure 3. ‘Red circles represent MRI abnormalities cited by a neuroradiologist, who was blinded to patient, condition, and other available imaging.’ Do you mean ‘red ellipses’? In addition, the ‘red ellipses’ can be more clear in the figure.
7. ‘Our preliminary results suggest that adult VA-ECMO patients with IABP support can be safely imaged with low-field POC brain MRI in the intensive care unit, allowing for the timely assessment and treatment of ABI.’ This study included only 3 patients; can the results from this small number of patients support such conclusion?
8. References. Ref. 1 is blank. Please check this.
Author Response
Reviewer 2
This study investigated the feasibility of bedside portable low-field brain MRI in extracorporeal membrane oxygenation (ECMO) patients supported on intra-aortic balloon pump. Three patients were involved.
- This study is actually a case report. Does this special issue of Diagnostics support such an article type?
Thank you for this question, we requested to submit our paper as a novel article rather than a case report so that we could share our pre-clinical safety data, methods, and safety information regarding IABP use near an MRI. Due to pre-planned pre-clinical phantom experiment, this is considered as an original research and we have communicated this with the Editors who agreed.
- The article type in Page 1 ‘Research Manuscript’ would be better to change to something like ‘Case Report.’
Thank you for your comment. We considered a case report/case series however as group we felt the pre-clinical data, methods, and safety considerations may be more important than the case report to institutions wishing to pursue this imaging modality in their ECMO patients. Reporting of these concepts and methods would be limited by a case report format and as such we requested permission to submit this as a novel article instead. We have communicated this with the Editors who agreed. Thank you very much.
- ‘Recent data suggests point-of-care (POC) magnetic resonance imaging (MRI) is safe and effective in diagnosing ABI in ECMO patients, no evidence exists regarding the safety of MRI for IABP patients.’ This sentence should be revised.
Thank you this sentence was revised to ‘Recent data suggests point-of-care (POC) magnetic resonance imaging (MRI) is safe and may be effective in diagnosing ABI in ECMO patients, however no data exists in patients supported on ECMO with an IABP’.
- There are ‘5G’ and ‘5-gauss’ and ‘5 Gauss’ throughout the manuscript. Please unify the descriptions. In addition, ‘5G’ may be mistaken to be understood as ‘5-Generation.’
Thank you for noticing this, we have changed the phrasing to 5-Gauss to remain uniform and avoid confusion with 5th Generation.
- ‘Impella devices’ would be better to change to ‘Impella® devices’?
Thank you, this has been updated.
- Figure 3. ‘Red circles represent MRI abnormalities cited by a neuroradiologist, who was blinded to patient, condition, and other available imaging.’ Do you mean ‘red ellipses’? In addition, the ‘red ellipses’ can be more clear in the figure.
Thank you for mentioning this – we have revised Figure 3 to improve imaging quality and organization of the figure.
- ‘Our preliminary results suggest that adult VA-ECMO patients with IABP support can be safely imaged with low-field POC brain MRI in the intensive care unit, allowing for the timely assessment and treatment of ABI.’ This study included only 3 patients; can the results from this small number of patients support such conclusion?
Thank you for pointing this out, the statement was revised with the words ‘and treatment’ removed. We hope this is the case but agree that with only 3 patients this is overstepping. Statement now reads ‘Our preliminary results suggest that adult VA-ECMO patients with IABP support can be safely imaged with low-field POC brain MRI in the intensive care unit, allowing for early and bedside imaging of patients’
- References. Ref. 1 is blank. Please check this.
Thank you for noticing. There was an editing error, the ‘1’ was erroneous and lead to all references being one number higher in the reference portion than intended, this has been corrected and the references have been re-arranged.
Round 2
Reviewer 2 Report
Thanks for the revision. My concerns have been addressed.